# Selenium Alleviates Cadmium Toxicity in Pepper (*Capsicum annuum* L.) by Reducing Accumulation, Enhancing Stress Resistance, and Promoting Growth

**DOI:** 10.3390/plants14091291

**Published:** 2025-04-24

**Authors:** Chen Cheng, Jianxiu Liu, Jiahui Liu, Zhiqiang Gao, Yang Yang, Bo Zhu, Fengxian Yao, Qing Ye

**Affiliations:** 1Life Sciences College, Gannan Normal University, Ganzhou 341000, China; chengchenzxm@163.com (C.C.); jianxiuliu200204@163.com (J.L.); jiahuiliu@163.com (J.L.); zhi_qiang_g@126.com (Z.G.); yyang_311@gnnu.edu.cn (Y.Y.); nczb615@163.com (B.Z.); fengxianyao@aliyun.com (F.Y.); 2South China Botanical Garden, Chinese Academy of Sciences, Guangzhou 510650, China

**Keywords:** heavy metal pollution, ion competition, selenium enrichment and cadmium reduction, toxicity reduction, endogenous hormone regulation

## Abstract

The enrichment of cadmium (Cd) is an important factor threatening crop growth and food safety. However, it is unclear whether exogenous selenium (Se) can simultaneously achieve Cd reduction and promote the growth of peppers. This study used Yuefeng 750 and Hongtianhu 101 as materials and investigated the interaction effects of different Se-Cd concentrations (Cd = 2 and 5 μM; Se = 0, 0.5, and 2 μM) on the uptake and transport of Cd and Se, resistance physiology, and growth and development of pepper seedlings in a hydroponic experiment. The organ Cd content was significantly increased in pepper seedlings, inhibiting their growth and aggravating their physiological stress under Cd application. However, the growth and photosynthetic capacity of peppers were promoted after Se application under Cd stress. The superoxide anion (O_2_^−^), hydrogen peroxide (H_2_O_2_), malondialdehyde (MDA), and abscisic acid (ABA) contents and indole-3-acetic acid oxidase (IAAO) activity in the leaves showed a significantly progressive decline, while the proline (Pro), ascorbic acid (ASA), and trans zeatin riboside (ZR) contents showed a significant rising trend. Thus, the growth, development, and dry matter accumulation of peppers were enhanced by reducing Cd stress. Meanwhile, the application of exogenous Se significantly improved the accumulation of Se in seedlings. In addition, compared to Hongtianhu 101, the Yuefeng 750 cultivars had a greater Cd and Se enrichment capacity. The cultivation of Cd-excluding cultivars combined with exogenous Se addition can be used as a recommended solution to reduce Cd toxicity and achieve Cd reduction and Se enrichment in peppers under Cd pollution.

## 1. Introduction

Cd is one of the most toxic non-essential trace elements to the human body [1]. In 2014, the *National Soil Pollution Survey Bulletin* issued by the Ministry of Environmental Protection and the Ministry of Land and Resources of the People’s Republic of China, showed that the rate of Cd exceedance in cultivated soil (7.0%) ranked first among heavy metals and metalloids, indicating that soil Cd pollution was the most serious in China [2]. Due to human activities such as mining, fertilization, and pesticide application, both the areas affected by heavy metal pollution and the severity of soil contamination continue to increase. Cd absorbed by crop roots is readily translocated to edible tissues, where it accumulates, posing significant human health risks through dietary exposure [3]. At the same time, Cd stress inhibits crop growth, leading to reduced crop yield or death [4]. Therefore, it is particularly important to reduce the accumulation of Cd in the edible parts of crops and promote crop growth in Cd-contaminated areas.

In recent years, many strategies have been reported to reduce the absorption of heavy metals and alleviate their toxicity in crops. On the one hand, there are strategies for removing heavy metals from contaminated soil, such as soil washing remediation or planting Cd-super-enriched plants on heavy metal-contaminated farmland, but the costs are higher and they may have adverse effects on soil quality and the ecological environment [5]. On the other hand, there are strategies for reducing Cd accumulation in crops by reducing the bioavailability of soil Cd, breeding low-accumulation varieties, and regulating water and fertilizer [6,7]. Among them, mineral elements (such as Se, magnesium, and silicon) play a key role in crop growth and have the ability to reduce the accumulation of Cd and other heavy metals in crops.

Se is an indispensable and important trace element in humans and animals, and Se deficiency can lead to many human diseases such as Keshan disease and Kashin–Beck disease [8]. Although Se is not an essential element for crop growth and development, Se has received extensive attention in reducing crop Cd accumulation and resisting heavy metal toxicity [9]. Se can enhance the antioxidant effect, remove excessive reactive oxygen free radicals, reduce lipid peroxidation, and reduce the damage caused by heavy metal-induced oxidative stress. It can also enhance crop growth and development by promoting root growth, photosynthetic system repair, and chloroplast reconstruction to reduce Cd toxicity [10,11]. Previous reports have shown that the protective effect of Se on crops mainly involves reducing the uptake and accumulation of Cd in some organs of crops or whole plants, such as rice [12], maize (corn) [13], tomato [14], and cilantro (coriander) [15]. Therefore, an appropriate amount of Se application can not only alleviate the toxicity of Cd and reduce the Cd content of crops but also increase the Se content of the edible parts to achieve safe Se supplementation for the human body.

Pepper (*Capsicum annuum* L.) is widely cultivated and consumed due to its high nutritional and medicinal value. Currently, the annual planting area of peppers in China remains above 2.1 million hm^2^ [16]. Compared to vegetables such as cabbage, roots, melons, and beans, solanaceous vegetables have higher accumulation of Cd and greater enrichment ability, especially peppers [17]. However, there are few studies on reducing Cd accumulation and toxicity in peppers by adding exogenous Se, and it is not clear whether the overall Se enrichment of crops can be achieved simultaneously. Se’s ability to improve pepper resistance to reduce Cd toxicity may be related to antioxidants, plant hormones, and related enzymes. Based on the protective mechanism of plant growth under Cd stress, we speculate that Se affects the stress resistance of peppers by regulating their endogenous hormone and antioxidant substance contents.

Taking two pepper varieties as the research objects, the effects of Se addition under different Cd stress levels on the uptake and transport of Cd and Se in pepper roots, as well as on resistance physiology, growth, and development under hydroponic conditions, were explored. We studied whether Cd toxicity can be reduced by reducing Cd absorption in peppers, increasing stress resistance and improving root and leaf growth and function. We examined whether it is possible to simultaneously achieve Cd reduction and Se enrichment in the edible parts of crops. The results provide a theoretical basis for the safe utilization of Cd-contaminated vegetable land and the simultaneous achievement of low-Cd, Se-rich, high-yield pepper cultivation.

## 2. Results

### 2.1. Characteristics of Cd and Se in Pepper Seedlings

Except for the leaves of Hongtianhu 101, an increase in Cd concentration in the nutrient solution significantly increased the Cd content in the roots, stems, and leaves of the two varieties of pepper seedlings, while the addition of Se to the nutrient solution significantly reduced the Cd content in each organ (roots, stems, and leaves) under different varieties and Cd concentrations (*p* < 0.05). Compared to Se0, the Cd content in the roots in the Se0.5 and Se2 treatments decreased by 12.7–26.4% and 24.8–60.6% (Figure 1A,B), in the stems by 14.6–7.5% and 25.1–40.7% (Figure 1C,D), and in the leaves by 11.9–32.1% and 27.5–41.5%, respectively (Figure 1E,F). There was no significant interaction between Cd and Se in the nutrient solution on the Cd content of each organ in Hongtianhu 101.

With an increase in Se concentration in the nutrient solution, the Se content in the roots, stems, and leaves of the pepper seedlings increased significantly, but a high concentration of Cd inhibited the absorption and transport of Se, and the regulation effect on the Se content in the stems of the two varieties reached a significant level (Appendix A). Compared to Cd2, the Se content in the roots, stems, and leaves of the same variety under Cd5 treatment decreased by 14.4–64.0%, 11.3–40.9, and 18.6–37.2%, respectively (excluding the Se0 treatment). Except for the stems of Hongtianhu 101, the Se content of each organ of the seedlings did not reach a significant level in the interaction of different Cd and Se concentrations of the nutrient solution.

From the perspective of the Cd and Se contents in each organ (roots, stems, and leaves) of the pepper plants, the levels followed the following order: roots > stems > leaves. The Cd and Se contents gradually decreased during the absorption and transport through the roots, stems, and leaves of the pepper plants (Figure 1 and Appendix A). The absorption and transport characteristics of Cd and Se in the different pepper varieties were quite different. Compared to Hongtianhu 101, the Cd contents in the roots, stems, and leaves of Yuefeng 750 increased by 81.3–143.6%, 11.2–33.6%, and 146.2–279.2%, and the Se contents increased by 128.0–294.4%, 13.3–111.5%, and 58.6–111.1%, respectively (excluding Se_0_ treatment).

### 2.2. Total Accumulation of Cd and Se in Pepper Seedlings

An increase in Cd concentration in the nutrient solution significantly increased the total Cd accumulation in the two varieties of pepper seedlings (*p* < 0.05), while an increase in Se concentration significantly reduced the accumulation of Cd. Compared to Se0, the total accumulation of Cd in the seedlings treated with Se0.5 and Se2 decreased by 4.7–10.8% and 17.9–37.8%, respectively (Figure 2A,B). However, the total accumulation of Cd in the seedlings did not reach a significant level in the interaction effect of Cd and Se in the nutrient solution.

The addition of Se to the nutrient solution significantly increased the accumulation of Se in the two varieties of pepper seedlings, but we found that an increase in Cd concentration in the nutrient solution significantly inhibited the absorption of Se. Compared to Cd2, the total accumulation of Se in the Cd5 treatment decreased by 29.0–54.4% (Figure 2C,D). Moreover, there was a significant interaction between the Cd and Se treatments in the nutrient solution on the total accumulation of Se in the seedlings.

From the perspective of cultivar characteristics, compared to Hongtianhu 101, the total Cd accumulation in the Yuefeng 750 seedlings increased by 191.4–210.9%, and the total Se accumulation increased by 166.2–299.1% (excluding Se0 treatment). In addition, it is also worth noting that under the same concentration of Se and Cd (Se2, Cd2) in the nutrient solution, the total amount of Cd absorbed by the pepper seedlings was 3.8–3.9 times that of Se, indicating that the migration performance of Cd in the nutrient solution–plant system is stronger than that of Se.

### 2.3. Resistance Physiology

#### 2.3.1. Analysis of O_2_^−^, H_2_O_2_, and MDA Contents

In addition to O_2_^−^ in the leaves of Hongtianhu 101, an increase in Cd concentration in the nutrient solution significantly increased the contents of ROS such as O_2_^−^, H_2_O_2_, and MDA in the leaves of the two varieties of pepper seedlings (Figure 3). Based on the addition of exogenous Se to the nutrient solution while under Cd stress, the production of ROS could be significantly reduced, the oxidative stress induced by Cd could be alleviated, and the degree of membrane lipid peroxidation could be significantly reduced. With an increase in Se concentration in the nutrient solution, the ROS (O_2_^−^, H_2_O_2_) and MDA contents in the pepper seedling leaves decreased significantly. The addition of Cd and Se to the nutrient solution resulted in no significant interaction with the O_2_^−^, H_2_O_2_, and MDA contents in the leaves of the two varieties. In addition, compared to Hongtianhu 101, the O_2_^−^, H_2_O_2_, and MDA contents in the leaves of the Yuefeng 750 seedlings increased by 28.1–31.3%, 3.2–35.7%, and 6.8–18.2%, respectively.

#### 2.3.2. An Analysis of the Pro and ASA Contents

With an increase in Cd concentration in the nutrient solution, the Pro content in the leaves of the pepper seedlings increased significantly, while the ASA content decreased significantly (Appendix A). Increasing the concentration of Se in the nutrient solution under Cd stress significantly increased the leaf Pro and ASA contents, which was conducive to improving the Cd tolerance of the pepper seedlings. The addition of Cd and Se to the nutrient solution resulted in no significant interaction effects on the leaves of the two varieties. In addition, compared to Hongtianhu 101, the Pro and ASA contents in the leaves of the Yuefeng 750 seedlings decreased by 2.8–8.6% and 3.8–21.6%, respectively.

#### 2.3.3. Analysis of Antioxidative Enzyme Activities: Superoxide Dismutase (SOD), Catalase (CAT), and Peroxidase (POD)

The response of antioxidant enzyme activities to Cd stress in the leaves of the pepper seedlings under the addition of Se and Cd to the nutrient solution differed (Appendix A). An increase in Cd concentration in the nutrient solution increased the SOD and POD activities in the leaves of the pepper seedlings, while the addition of Se significantly reduced this increase based on Cd stress. However, an increase in Cd concentration decreased the CAT activity of the seedling leaves, but the effect of Cd on CAT activity was significantly offset by the addition of Se to the nutrient solution. There was no significant interaction between Cd and Se on the CAT, SOD, and POD activities in the leaves of the two varieties of pepper seedlings. In addition, compared to Hongtianhu 101, the SOD and POD activities in Yuefeng 750 increased by 13.6–34.2% and 6.4–13.2%, respectively, while the CAT activity decreased by 3.7–18.7%.

#### 2.3.4. An Analysis of the ABA and ZR Contents and the IAAO Activity

Cd stress in the nutrient solution led to an increase in the ABA content and a significant decrease in the ZR content in the leaves of the pepper seedlings, while the addition of Se under Cd stress significantly reduced the ABA content and increased the ZR content (Figure 4). At the same time, Cd stress also affected the IAA content of the leaves by regulating the IAAO activity; if the IAAO activity of the plants increased, the IAA content decreased, and vice versa. Figure 4 shows that the IAAO activity of the leaves under Cd stress in the nutrient solution increased significantly, while the addition of Se, based on Cd stress, significantly reduced the IAAO activity. There was no significant interaction between Cd and Se observed in terms of the ABA and ZR contents or the IAAO activity in the leaves of the two varieties of pepper seedlings. In addition, compared to Hongtianhu 101, the ABA content and IAA activity of Yuefeng 750 increased by 6.7–14.4% and 23.8–123.7%, respectively, while the ZR content decreased by 30.8–41.7%.

### 2.4. Changes in the Characteristics of Roots

An increase in Cd concentration in the nutrient solution significantly reduced the root length, root surface area, root volume, root activity, and root tip number of the two varieties of pepper seedlings but significantly increased the root diameter. The addition of exogenous Se to the nutrient solution significantly reduced the adverse effects of Cd stress on the roots (Appendix A). Compared to Hongtianhu 101, the root length, surface area, volume, diameter, and root tip number of Yuefeng 750 significantly increased by 1.6–67.7%, 7.9–26.4%, 2.4–5.0%, 1.3–10.8%, and 6.5–28.2%, respectively, but the root activity decreased by 28.2–72.3%. In addition to root length, there were no significant interactions between Cd, Se, and pepper varieties in the nutrient solution on the root surface area, volume, diameter, root tip number, and root activity.

### 2.5. Changes in the Characteristics of Leaves

An increase in Cd concentration in the nutrient solution significantly reduced the leaf area and photosynthetic pigment contents in the two varieties of pepper seedlings, while the addition of Se under Cd stress significantly increased the total leaf area and chlorophyll a, chlorophyll b, and carotenoid contents of the seedlings, which was conducive to reducing the adverse effects of Cd stress (Table 1). Compared to Hongtianhu 101, the leaf area of Yuefeng 750 increased significantly by 1.7–33.1%, but the chlorophyll a, chlorophyll b, and carotenoid contents decreased significantly by 2.0–8.7%, 8.8–22.4%, and 2.3–5.3%, respectively. The addition of Cd and Se to the nutrient solution, as well as pepper varieties, resulted in no significant interaction effects on the leaf area or chlorophyll a, chlorophyll b, and carotenoid contents of the seedlings.

### 2.6. Plant Type

The addition of Cd to the nutrient solution significantly reduced the plant height, stem base thickness, and canopy width of the pepper seedlings (Appendix A). With an increase in Se concentration in the nutrient solution under Cd stress, the plant height, stem base diameter, and canopy width of the pepper seedlings increased significantly for the different varieties under different Cd concentrations. Compared to Hongtianhu 101, the plant height, stem base diameter, and canopy width of Yuefeng 750 increased significantly by 32.3–74.7%, 15.3–28.2%, and 6.8–28.4%, respectively. In addition to the stem base width, Cd and Se addition to the nutrient solution resulted in significant interaction effects on the seedling plant height and canopy width.

### 2.7. Biomass Production

With an increase in Cd concentration in the nutrient solution, the dry weight of the roots, stems, leaves, and other organs of the two varieties of pepper seedlings decreased significantly. The growth inhibition effect induced by Cd was significantly reduced after the application of Se. Compared to Se0, the dry weight of the roots induced by Se0.5 and Se2 significantly increased by 8.4–23.6% and 15.5–44.9% (Figure 5A,B), the stem dry weight increased by 10.2–14.6% and 18.7–52.2% (Figure 5C,D), and the leaf dry weight increased by 6.6–29.9% and 11.8%–40.5%, respectively (Figure 5E,F). The addition of Cd and Se to the nutrient solution resulted in no significant interaction effects on the dry weight of the roots, stems, or leaves of the two varieties of pepper seedlings. In addition, compared to Hongtianhu 101, the dry weight of the roots, stems, and leaves of Yuefeng 750 increased by 16.2–59.8%, 73.9–117.3%, and 16.0–45.7%, respectively.

## 3. Discussion

Our results demonstrate that Se application significantly inhibits Cd uptake and accumulation in pepper plants. Shanker et al.’s study [9] supports this result, showing an inverse correlation between selenite concentration and Cd uptake in both root and shoot tissues. Similar results have also been reported by Wan et al. [18] and Xie et al. [19]. The regulatory mechanisms behind this phenomenon may involve the following three aspects: First, a possible reason is the formation mechanism of Cd–Se complexes in the roots, which reduces the amount of bioavailable Cd. In plant root cells, different oxidation states of Se (such as Se (IV) and Se (VI)) can be reduced to Se^2−^, which combines with Cd ions to form insoluble Cd–Se complexes [9], thereby reducing crop uptake of Cd. Wang et al. [20] further confirmed the presence of these complexes in *Pseudomonas aeruginosa*. The formation of Cd–Se complexes not only reduces the solubility of Cd and prevents its free migration but may also alter the absorption kinetics of Cd in plant roots. Second, the plant cell wall has enormous mechanical rigidity and strength and can be regarded as the first barrier for heavy metal ions to enter crop cells. Exogenous Se increases the pectin and hemicellulose 2 (HC2) contents in the root cell wall and fixes metal Cd to the cell wall [21]. Meanwhile, the application of exogenous Se can increase the contents of chelating peptides (PCs), glutathione (GSH), and metallothionein (MT) in crops. Cd combines with these substances to form Cd protein chelates and immobilizes them in crop organs, inhibiting the transfer of Cd to other crop organs and thereby reducing the concentration of free metal Cd ions. In particular, in this study, the Cd and Se contents decreased gradually during absorption and transport in pepper plants, showing a root > stem > leaf order (Figure 1). This rule may be related to the fact that Se can reduce the transfer coefficient of Cd, inhibit the transfer of Cd from the roots to the aboveground parts, and reduce the accumulation of Cd in the in aboveground parts [22].

When plants are subjected to Cd stress, the excessive accumulation of reactive oxygen species (ROS) in the body leads to oxidative stress, which results in the peroxidation of important cell structures, such as lipids and proteins, thus limiting normal metabolic efficiency [23]. In this study, we observed that Se treatment significantly counteracted the increase in H_2_O_2_ and O_2_^−^ levels induced by Cd (Figure 3), showing that Se can reduce oxidative stress, which is consistent with the result of Zhang et al. [24].

To avoid ROS damage, plants have formed an enzymatic ROS scavenging mechanism in long-term adaptive stress [25]. First, SOD converts ROS such as O_2_^−^ in plants into H_2_O_2_. Second, POD and CAT further convert them into completely non-toxic O_2_^−^ and H_2_O, thereby reducing the accumulation of ROS [26]. Lin et al. [12] believed that the significant decrease in Cd stress that they observed in rice seedlings under Se treatment was mainly attributed to the fact that Se application decreased the Cd-induced increase in SOD and POD activities but elevated depressed CAT activity. This result is consistent with the present experiment. The decrease in POD and SOD activities under Se treatment may be due to the significant increase in CAT activity on the one hand and the removal of some oxidized free radicals in non-enzymatic form on the other hand [26]. Non-enzymatic antioxidants are able to scavenge free radicals by breaking the free radical chain reaction. In this study, Se enhanced the increase in ASA levels caused by Cd, indicating that the pepper seedlings tried to cope with the Cd-induced oxidative stress by strengthening their antioxidant capabilities. However, Saidi et al. [27] observed that Se caused a reduction in leaf ASA content in sunflower seedlings under Cd stress. The ASA content in the two seedling types showed an opposite trend change, which may have been caused by the difference in the cycle efficiency of ASA oxidation forms in different plants. Furthermore, some studies have found that Se can increase the content of glutathione (GSH) [28], vitamin C, tocopherol [29], flavonoids, and acid-soluble thiols in plant tissues to resist Cd stress. Nevertheless, more information is needed to determine the contribution of Se to the antioxidative system of Cd toxicity in pepper. Seppänen et al. [30] discovered that Se has synergistic effects on the transcription of antioxidative enzymes such as Cu, Zn, SOD, and GPX in plants. Se promotes H_2_O_2_ scavenging by increasing GPX activity, which has been initially identified as an abiotic stress-responsive enzyme [31]. In the current study, we also observed that Se addition continued to increase the Pro content induced by Cd stress. Pro can measure the resistance of plants to a certain extent, and the higher the proline content, the stronger the resistance [32]. This indicates that Se can increase the accumulation of osmotic protective agents such as Pro in pepper, provide additional antioxidant support for the plants, and maintain the osmotic balance in cells, thereby improving plant resistance to Cd.

It has been shown that in addition to regulating plant development in a normal environment, plant hormones can also regulate plant growth adaptability in response to various environmental stresses. In this study, it was found that the ABA content in the leaves of pepper seedlings increased under Cd stress, while the zeatin nucleoside (ZR) and indole 3-acetic acid (IAA) contents decreased significantly, and plants could regulate the synthesis of various hormones, signal transduction, and metabolism in response to Cd stress through a variety of crosstalk, thereby constructing a defense system [33]. Another study showed that the administration of Se further increased the expression level of the signal transduction genes involved in stress-related plant hormones [34]. Se may alter this stress response and balance the antioxidant system, which helps to slow down the negative effects of oxidative stress on plants, leading to lower ABA levels. In addition, Se achieved a complex effect on the root meristem through an auxin–cytokinin antagonism, and the results showed that the ZR and IAA contents could be increased by adding Se in an appropriate amount. When plants are subjected to heavy metal stress, the cytokinin activated by Se-treated plants may lead to NO production, which can induce their defense mechanisms against stress [35]. Second, CK regulates the expression of non-protein thiol genes and regulates the synthesis of detoxification substances in plants. We also found that Se addition significantly reduced IAAO activity under Cd stress. IAAO can catalyze the conversion of IAA to OxIAA. Therefore, we infer that the addition of Se leads to less decomposition of IAA and an increase in IAA content. The analysis of auxin-related genes by Luo et al. [36] showed that the application of lower concentrations of Se increased the expression levels of YUCCA and NtPIN family genes, indicating that Se could affect root growth by increasing auxin concentrations under Cd stress, which is similar to our experimental results.

The level of MDA, a product of lipid peroxidation, is often used as an indicator of the degree of oxidative damage to cell membranes [37]. This study confirmed that Se supplementation under Cd stress significantly inhibits the increase in MDA caused by Cd, indicating that the concentration of exogenous Se in this experiment could alleviate the lipid peroxidation of pepper seedlings under Cd stress. Similar results have been obtained for broccoli (*Brassica oleracea*) plants [38], rape and wheat seedlings [39], and sunflower (*Helianthus annuus*) seedlings [27]. Based on our results, this protective effect of Se may be related to the improvement of the ROS scavenging capability of non-enzymatic and enzymatic antioxidants, thereby reducing the ROS content, optimizing hormonal homeostasis and relieving membrane lipid peroxidation.

In summary, the present study provides compelling evidence that Cd stress significantly increases the accumulation of Cd in peppers and inhibits the growth and development of pepper seedlings. Remarkably, Se application counteracted Cd stress in pepper seedlings, resulting in enhanced biomass production across all organs (roots, stems, and leaves) and significant improvements in key growth traits (plant height, stem diameter, and canopy width). Previous studies have indicated that the presence of Se stimulates plant production and dry biomass accumulation of bell peppers [40]. Huang et al. [41] also reconfirmed this view and demonstrated that the mean biomass of rice increased by 16.6% (11.4–22.3%) with Se addition under Cd-stressed environments in hydroponic experiments via a meta-analysis, while the shoot and root biomass of rice increased by 14.5% (8.44–21.2%) and 21.0% (11.4–31.5%), respectively. The Se-mediated alleviation of Cd toxicity could be explained by the following three mechanisms: First, Se supplementation under Cd stress enhanced the root vigor and photosynthetic performance of pepper seedlings through coordinated physiological improvements. In the root system, the addition of exogenous Se significantly increased the root length, root surface area, root volume, root tip number, and root viability of the two varieties of pepper seedlings that were inhibited by Cd stress (Appendix A). Sun et al. [42] showed that Se can accelerate the mitosis of garlic root apical cells under Cd stress. Moreover, exogenous Se application reduced the proportion of fine roots and increased the proportion of medium roots [43]. These findings demonstrate that Se confers protection against Cd-induced root growth inhibition through morphological optimization and cellular activity maintenance. Concurrently, Se addition under Cd stress significantly increased the total leaf area and chlorophyll a, chlorophyll b, and carotenoid contents of the seedlings (Table 1). Huang et al. [41] also reported that the chlorophyll content of rice was increased by 18.6% (14.7–22.6%) with Se supplemented under Cd stress. Se can reduce the chloroplast dysfunction caused by Cd stress by reconstructing the chloroplast ultrastructure and thylakoid and matrix structure and increasing the plant chlorophyll content [44]. Therefore, Se significantly enhances the photosynthetic capacity of pepper seedlings under Cd stress by maintaining the structural and functional stability of chloroplasts [45]. Second, under Cd stress, Se addition significantly reduced the Cd content and total accumulation in various organs of the two pepper varieties (Figure 1 and Figure 2), reducing the inhibitory effect of Cd on crops [46]. Third, Se treatment significantly reduced the degree of membrane lipid peroxidation in the seedlings by regulating the antioxidant enzyme activity (CAT, POD, and SOD), antioxidant content (Pro and ASA), and plant hormone system (ABA, IAA, and ZR), thereby enhancing the plant’s resistance to Cd stress and promoting the normal growth and development of pepper plants (Figure 4 and Appendix A). Therefore, our analysis revealed that Se alleviates Cd toxicity through a tripartite mechanism: (1) promoting plant growth and development, (2) inhibiting Cd translocation, (3) enhancing stress resistance. These findings show that Se provides a feasible solution for the high-quality and safe production of crops in Cd-contaminated soil.

## 4. Materials and Methods

### 4.1. Experimental Materials

The Cd-high accumulation variety Yuefeng 750 (Jiangxi Yuefeng Seed Co., Ltd., Nanchang, China) and the Cd-low accumulation variety Hongtianhu 101 (Sannis Seed (Beijing) Co., Ltd., Longmont, CO, USA) were selected as the tested pepper varieties. Na_2_SeO_3_ (Xilong Science Co., Ltd., Guanzhou, China) had a purity (Na_2_SeO_3_) ≥ 97.0%, and CdCl_2_ was anhydrous Cd chloride with 99% purity (Shanghai McLin Biochemical Technology Co., Ltd., Shanghai, China).

### 4.2. Experimental Design

Pepper seedling cultivation was conducted as the initial step of the experiment. On 29 October 2022, the seeds of the test peppers were sterilized with 0.2% KMnO_4_ solution for 20 min, rinsed with distilled water many times, soaked in water for 24 h, rinsed several times, sown in a 72-hole tray for seedlings, and placed in a plant culture room for growth. The light cycle of the plant culture room was 16 h, and the temperature was controlled at 20/26 °C (night/day).

Subsequently, pepper seedling transplantation and hydroponic cultivation were conducted. After sowing for 20 days, healthy and uniformly growing pepper seedlings (1–2 true leaves) were transplanted into a white rectangular sponge board (a single sponge board was perforated in 4 rows and 3 columns, with a total of 12 holes) and placed in a black plastic container (length, width, and height of 35 × 27 × 13 cm) containing 5 L of half-strength Hoagland modified nutrient solution (the size of the sponge board was consistent with the diameter of the black plastic container). After 4 days, the nutrient solution was replaced with a full-strength Hoagland nutrient solution. The Hoagland nutrient solution was composed of working solution A (4 mmol L^−1^ of Ca(NO_3_)_2_·4H_2_O and 6 mmol L^−1^ of KNO_3_), working solution B (2 mmol L^−1^ of MgSO_4_·7H_2_O and 1 mmol L^−1^ of NH_4_H_2_PO_4_), and working solution C (80 μmol L^−1^ of NaFe-EDTA, 46.3 μmol L^−1^ of H_3_BO_3_, 9.5 μmol L^−1^ of MnSO_4_·H_2_O, 0.8 μmol L^−1^ of ZnSO_4_·7H_2_O, 0.3 μmol L^−1^ of CuSO_4_, and 0.02 μmol L^−1^ of (NH_4_)_6_Mo_7_O_24_). The pH of the solution was adjusted to 6.2 ± 0.1 using 1.0 mmol L^−1^ of NaOH or HCl, and the nutrient solution was changed every 2 days.

Experimental treatments were conducted from 4 to 30 days after transplanting the pepper seedlings into the hydroponic system. Cd (2 and 5 μmol L^−1^) and Se (0, 0.5, and 2 μmol L^−1^) were added to the Hoagland nutrient solution in the form of CdCl_2_ and Na_2_SeO_3_, respectively. A two-factor completely random design (recorded as Cd2 Se0, Cd2 Se0.5, Cd_2_ Se2, Cd5 Se0, Cd5 Se0.5, and Cd5 Se2) was adopted, with a total of 6 treatments and 3 replicates. The trial ended on 19 December 2022, when the plants were harvested. Representative plants were selected for each treatment. The samples were first rinsed with tap water, and then the roots were immersed in 20 mmol L^−1^ of Na_2_-EDTA solution for 20 min to remove the Cd and Se ions adsorbed on the root surface. Finally, they were washed with deionized water and wiped dry to determine the following indicators.

### 4.3. Plant Sampling and Analysis

#### 4.3.1. Determination of Pepper Plant Type

Plant height and canopy width were measured with a ruler, and stem base width was measured using a digital vernier caliper (China Ningbo Deli Information Technology Co., Ltd., Ningbo, China; accuracy: ±0.2 mm; resolution: 0.1 mm) following standardized protocols. The root and leaf types were determined and analyzed using an LA-S plant image analyzer system (Hangzhou Wanshen Detection Technology Co., Ltd., Hangzhou, China). Among them, the root analysis recorded the morphological parameters such as root length, surface area, root volume, average diameter and root tip number using the root analyzer system, and the leaf area measurement system analyzed the total leaf area of three representative seedlings, with three independent biological replicates analyzed per treatment.

#### 4.3.2. Determination of Root Activity and Chloroplast Pigment Content

The root activity of the peppers was determined using the 2,3,5-triphenyltetrazolium chloride (TTC) method as described by Yamauchi et al. [47], with modifications: Root tip samples (0.2–0.3 g) were measured, and the TTC reduction amount was calculated to assess the root dehydrogenase activity. The content of photosynthetic pigments was determined using Zhao et al.’s method [21]. The surface of the fresh pepper leaves was wiped clean, and then they were cut into pieces without the leaf midrib. Next, the leaves were mixed well, and 0.2 g was weighed out. Using 95% ethanol as the organic solvent, the absorbance was measured at the wavelengths of 665, 649, and 470 nm using a UV–Vis spectrophotometer (UV-300, Shanghai Mepida Instrument Co., Ltd., Shanghai, China), and the contents of leaf pigments (chlorophyll a, chlorophyll b, and carotenoids) were calculated.

#### 4.3.3. Dry Matter Production

For each treatment, six seedlings were separated into roots, stems, and leaves. The corresponding tissues from all seedlings were pooled to form one composite sample per plant part, with three independent replicate samples per treatment. The samples were first deactivated at 105 °C for 30 min, and then dried to a constant weight at 75 °C before weighing.

#### 4.3.4. Determination of the Total Se and Cd Contents in the Plants

The plant samples were ground into powder and sieved using a 100-mesh screen before use. Each sample (approximately 200 mg) was digested with 8 mL of HNO_3_ (Guaranteed Reagent, GR) and 2 mL of H_2_O_2_ (GR) using a MASTER-40 microwave digester (Shanghai Xinyi Microwave Chemical Technology Co., Ltd., Shanghai, China) for 30 min. After cooling the digestion tank and rinsing the inner cover with a minimal volume of distilled water, the digestate was then quantitatively transferred to a 50 mL volumetric flask and brought to volume with distilled water. After sample digestion, the total Cd and Se contents in the plant samples were determined using inductively coupled plasma mass spectrometry (Agilent 7900, Agilent Technology Co., Ltd., Santa Clara, CA, USA). The accuracy of the Se and Cd contents in the samples was verified using the element storage solution of the National Center of Standard Materials in China.

#### 4.3.5. Resistance Physiological-Related Indicators

Representative leaves were randomly selected for each treatment, and the surface of the leaves was rinsed with distilled water, then dried and wrapped with tin foil paper. The leaves were quickly cooled in liquid nitrogen for 2 min and then stored in an ultra-low temperature refrigerator (−80 °C) for later use. The ROS (H_2_O_2_, O_2_^−^), MDA, Pro, antioxidant enzyme (CAT, POD, and SOD), antioxidant substance (ASA/Vitamin C), and IAAO contents were determined with a kit (Suzhou Keming Biotechnology Co., Ltd., Suzhou, China) using ultraviolet–visible spectrophotometry, according to the manufacturer’s standardized protocols. Each treatment was measured three times.

The contents of the endogenous hormones ZR (a kind of cytokinin) and ABA in the leaves were determined using high-performance liquid chromatography (LC-20A, Shimadzu, Kyoto, Japan). Approximately 0.2 g of fresh leaves was weighed and put into 2 mL tubes, and 1 mL of 80% pre-cooled methanol was added to each tube, followed by homogenization at 4 °C for 16 h using an automatic homogenizer and then centrifugation at 8000× *g* for 10 min to obtain the supernatant. The residue was re-extracted with 80% methanol for 2 h, and the supernatants were combined. The organic phase in the supernatant was removed by blowing nitrogen at 40 °C, and petroleum ether was added for decolorization by extraction 3 times. After the upper ether phase was discarded, the pH was adjusted to 2.8 with saturated citric acid, extracted 3 times using sethyl acetate, and combined with the organic phase. It was then blown dry using nitrogen, dissolved in methanol, and filtered using a needle filter before testing. Samples (10 μL) were chromatographed on a Kromasil C18 reversed-phase column (250 × 4.6 mm). The ZR and ABA contents were calculated by measuring the peak area of the standard samples and the samples to be tested after adjusting the mobile phase configuration (1% acetic acid aqueous solution and methanol), flow rate (0.8 mL/min), and column temperature (30 °C). The UV detection wavelength was set at 254 nm, and the sampling duration was 35 min. The ABA and ZR standards were purchased from Shanghai Yuanye Biotechnology Co., Ltd., Shanghai, China. Three independent biological replicates were analyzed for each sample.

### 4.4. Statistical Analyses

All data are presented as the mean ± standard deviation of three biological replicates, which were statistically analyzed and processed using SPSS 22.0 software (IBM Research, New York, NY, USA) and Microsoft Excel 2021, and graphed using OriginPro 9.0 (OriginLab Corporation, Northampton, MA, USA). The results were analyzed using two- and three-way ANOVAs. The mean values of each treatment were subjected to multiple comparisons using the LSD test (*p* < 0.05).

## 5. Conclusions

Hydroponic experiments demonstrated that Cd stress significantly increased Cd accumulation and inhibited the growth of pepper seedlings. Se supplementation effectively alleviated Cd toxicity through three mechanisms: reducing Cd accumulation in seedlings, enhancing stress resistance, and promoting plant growth and development. Among them, selenium enhanced stress resistance by scavenging ROS (O_2_^−^ and H_2_O_2_), increasing osmoprotectants (Pro) and antioxidants (CAT and ASA), optimizing hormonal homeostasis (↓ABA, IAAO; ↑ZR), and suppressing lipid peroxidation (MDA). Additionally, the synergistic enhancement of root system functionality and leaf photosynthetic performance served as the fundamental driver for improved plant biomass accumulation. In particular, selenium addition reduced Cd accumulation while enhancing Se enrichment in pepper seedlings, providing an effective strategy for high-yield, safe, and high-quality pepper production in Cd-contaminated areas.

## Figures and Tables

**Figure 1 plants-14-01291-f001:**
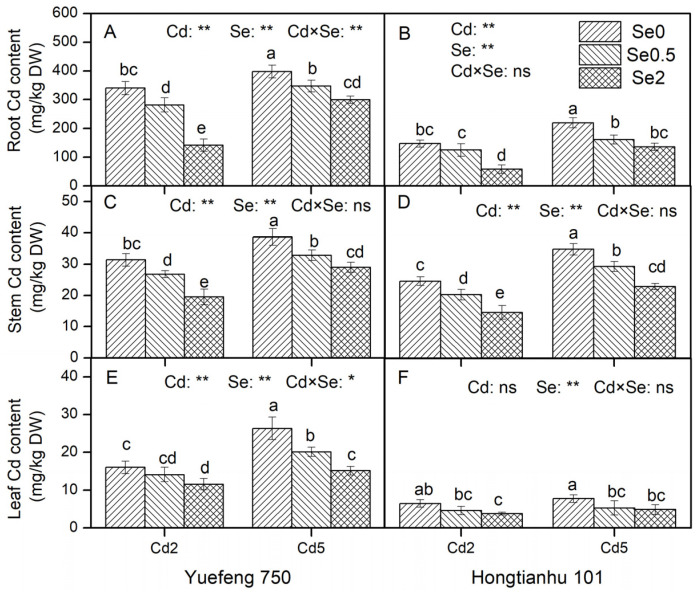
The effects of adding Cd and Se to the nutrient solution on the Cd content in the roots, stems, and leaves of the two varieties of pepper seedlings. Cd2 and Cd5 represent Cd concentrations of 2 and 5 μmol/L in the nutrient solution, while Se0, Se0.5, and Se2 represent Se concentrations of 0, 0.5, and 2 μmol/L, respectively. Changes in Cd content in the roots (**A**,**B**), stems (**C**,**D**), and leaf (**E**,**F**) of Yuefeng 750 and Hongtianhu 101 under combined Cd + Se treatment. Mean values (±SD, *n* = 3) followed by different lowercase letters indicate significant differences among the different treatments (*p* < 0.05) for the same variety according to a two-way ANOVA followed by an LSD test. Significant main and interactive effects are indicated by * (*p* < 0.05) and ** (*p* < 0.01); ns indicates non-significant effects.

**Figure 2 plants-14-01291-f002:**
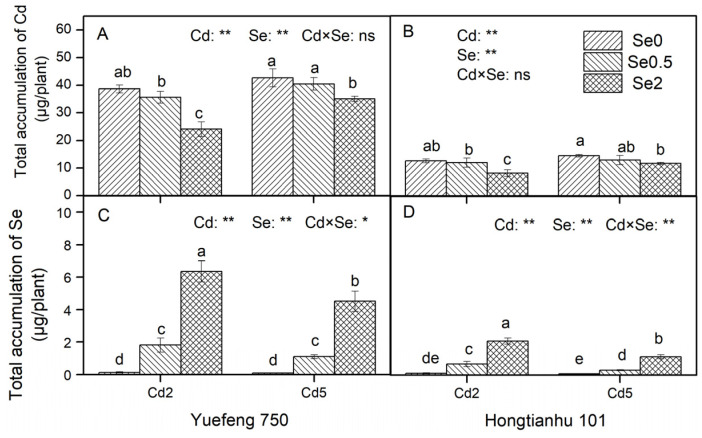
The effects of adding Cd and Se to the nutrient solution on the total Cd and Se accumulation in the two varieties of pepper seedlings. Cd2 and Cd5 represent Cd concentrations of 2 and 5 μmol/L in the nutrient solution, while Se0, Se0.5, and Se2 represent Se concentrations of 0, 0.5, and 2 μmol/L, respectively. Changes in total Cd accumulation (**A**,**B**), and total Se accumulation (**C**,**D**) in Yuefeng 750 and Hongtianhu 101 under combined Se+Cd treatment. Mean values (±SD, *n* = 3) followed by different lowercase letters indicate significant differences among the different treatments (*p* < 0.05) for the same variety according to a two-way ANOVA followed by an LSD test. Significant main and interactive effects are indicated by * (*p* < 0.05) and ** (*p* < 0.01); ns indicates non-significant effects.

**Figure 3 plants-14-01291-f003:**
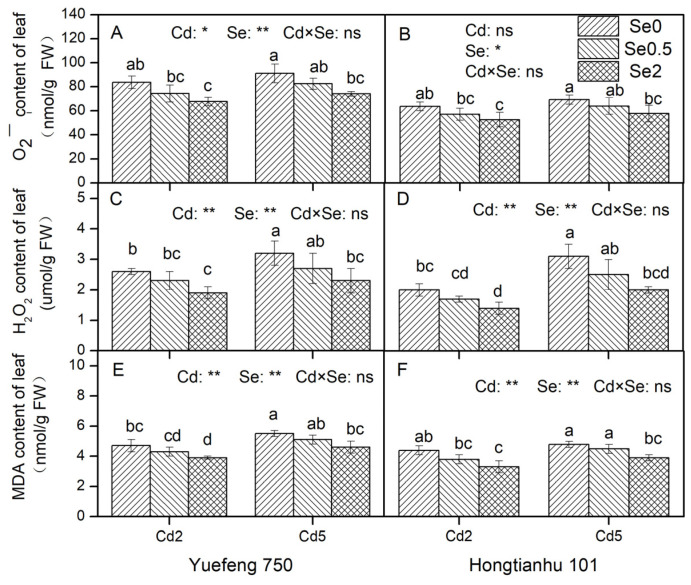
The effects of adding Cd and Se to the nutrient solution on the O_2_^−^, H_2_O_2_, and MDA contents in the leaves of the two varieties of pepper seedlings. Cd2 and Cd5 represent Cd concentrations of 2 and 5 μmol/L in the nutrient solution, while Se0, Se0.5, and Se2 represent Se concentrations of 0, 0.5, and 2 μmol/L, respectively. Changes in the leaf O_2_^−^ (**A**,**B**), H_2_O_2_ (**C**,**D**), and MDA (**E**,**F**) contents of in Yuefeng 750 and Hongtianhu 101 under combined Se+Cd treatment. Mean values (±SD, *n* = 3) followed by different lowercase letters indicate significant differences among the different treatments (*p* < 0.05) for the same variety according to a two-way ANOVA followed by an LSD test. Significant main and interactive effects are indicated by * (*p* < 0.05) and ** (*p* < 0.01); ns indicates non-significant effects.

**Figure 4 plants-14-01291-f004:**
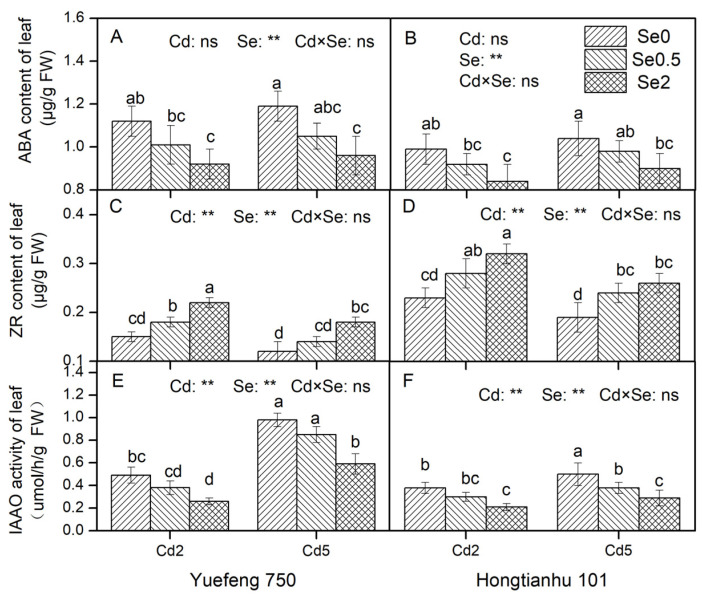
The effects of adding Cd and Se to the nutrient solution on the ABA and ZR contents and the IAAO activity of the leaves of the two varieties of pepper seedlings. Cd2 and Cd5 represent Cd concentrations of 2 and 5 μM in the nutrient solution, while Se0, Se0.5, and Se2 represent Se concentrations of 0, 0.5, and 2 μM, respectively. Changes in the leaf ABA contents (**A**,**B**), ZR contents (**C**,**D**), and IAAO Activity (**E**,**F**) of Yuefeng 750 and Hongtianhu 101 under combined Se + Cd treatment. Mean values (±SD, *n* = 3) followed by different lowercase letters indicate significant differences among the different treatments (*p* < 0.05) for the same variety according to a two-way ANOVA followed by an LSD test. Significant main and interactive effects are indicated by ** (*p* < 0.01); ns indicates non-significant effects.

**Figure 5 plants-14-01291-f005:**
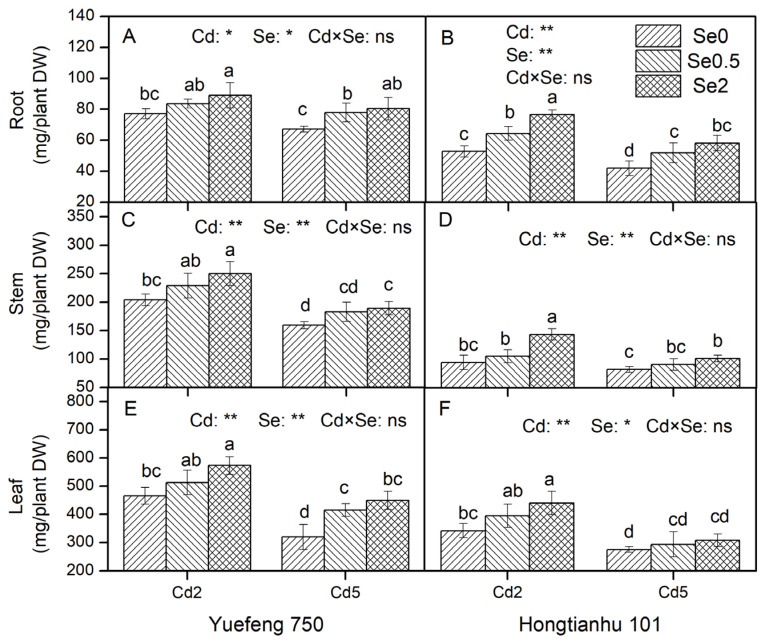
The effects of adding Cd and Se to the nutrient solution on the dry weight of the roots, stems, and leaves of the two varieties of pepper seedlings. Cd2 and Cd5 represent Cd concentrations of 2 and 5 μM in the nutrient solution, while Se0, Se0.5, and Se2 represent Se concentrations of 0, 0.5, and 2 μM in the nutrient solution, respectively. Changes in the dry weight of the roots (**A**,**B**), stems (**C**,**D**), and leaf (**E**,**F**) of Yuefeng 750 and Hongtianhu 101 under combined Cd + Se treatment. The data reflect the means of three independent replicates, and each replicate contained four plants. Mean values (±SD) followed by different lowercase letters indicate significant differences among the different treatments (*p* < 0.05) for the same variety according to a two-way ANOVA followed by an LSD test. Significant main and interactive effects are indicated by * (*p* < 0.05) and ** (*p* < 0.01); ns indicates non-significant effects.

**Table 1 plants-14-01291-t001:** The effects of adding Cd and Se to the nutrient solution on the leaf area and photosynthetic pigments of the two varieties of pepper seedlings.

Variety	Treatment	Leaf Area (cm^2^/Plant)	Chlorophyll a (mg/g FW)	Chlorophyll b (mg/g FW)	Carotenoids (mg/g FW)
Yuefeng 750	Cd2 Se0	187.4 ± 13.1 bc	1.91 ± 0.11 bc	0.53 ± 0.04 bc	0.42 ± 0.03 cd
	Cd2 Se0.5	215.1 ± 16.3 ab	2.06 ± 0.14 ab	0.58 ± 0.07 ab	0.44 ± 0.02 bc
	Cd2 Se2	228.6 ± 15.0 a	2.27 ± 0.17 a	0.63 ± 0.04 a	0.51 ± 0.04 a
	Cd5 Se0	158.0 ± 9.3 c	1.77 ± 0.11 c	0.47 ± 0.03 c	0.39 ± 0.02 d
	Cd5 Se0.5	186.1 ± 19.2 bc	1.90 ± 0.07 bc	0.52 ± 0.04 bc	0.43 ± 0.03 bcd
	Cd5 Se2	204.7 ± 23.8 ab	2.12 ± 0.11 ab	0.59 ± 0.04 ab	0.49 ± 0.03 ab
Hongtianhu 101	Cd2 Se0	143.8 ± 5.3 c	2.09 ± 0.05 bc	0.63 ± 0.03 bc	0.44 ± 0.02 cd
	Cd2 Se0.5	193.3 ± 13.0 b	2.18 ± 0.15 ab	0.68 ± 0.02 ab	0.47 ± 0.02 bc
	Cd2 Se2	224.8 ± 15.6 a	2.41 ± 0.17 a	0.69 ± 0.05 a	0.52 ± 0.03 a
	Cd5 Se0	118.7 ± 10.2 d	1.87 ± 0.12 c	0.61 ± 0.01 c	0.41 ± 0.04 d
	Cd5 Se0.5	147.4 ± 11.7 c	2.04 ± 0.10 bc	0.65 ± 0.01 abc	0.45 ± 0.02 bcd
	Cd5 Se2	159.6 ± 14.1 c	2.17 ± 0.14 ab	0.67 ± 0.05 ab	0.50 ± 0.03 ab
Statistical significance					
Variety (V)		**	**	**	*
Cd		**	**	*	*
Se		**	**	**	**
V × Cd		ns	ns	ns	ns
V × Se		ns	ns	ns	ns
Cd × Se		ns	ns	ns	ns
V × Cd × Se		ns	ns	ns	ns

Note: The pepper seedlings were grown in a nutrient solution with Cd (2 and 5 μM) and/or Se (0, 0.5, and 2 μM) for 26 days. The mean values (±SD, *n* = 3) in each column followed by different lowercase letters indicate significant differences among the different treatments (*p* < 0.05) for the same variety according to a three-way ANOVA followed by an LSD test. Significant main and interactive effects are indicated by * (*p* < 0.05) and ** (*p* < 0.01); ns indicates non-significant effects.

## Data Availability

Data are contained within the article.

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
