# Peer review of "Selenium Alleviates Cadmium Toxicity in Pepper (Capsicum annuum L.) by Reducing Accumulation, Enhancing Stress Resistance, and Promoting Growth"

_plants, 2025, doi:10.3390/plants14091291_

Round 1

Reviewer 1 Report

Comments and Suggestions for Authors

The main objective of this article is the evaluation of whether exogenous application of selenium (Se) can reduce cadmium (Cd) toxicity in plants of two pepper varieties (one with high and one with low Cd accumulation), grown in a hydroponic system. This is interesting because Cd is a problem in many agricultural lands and the use of Se to mitigate these effects is innovative.

Thus, this ms can be published in this journal after reviewing the next points. In addition to correcting the English, as there are several grammatical errors.

  • Line 1: Improve the title, the word Cadmium appears too often in the title.
  • Line 34: Is the most recent regulation from 2014?
  • Line 39: Rewrite ‘Cad is easily transported and accumulates...’.
  • Line 64: Unify the naming style of the listed plant species. While most are mentioned by their common name, Coriandrum sativum appears in scientific nomenclature.
  • Line 89: Why do the results appear before the materials and methods? Section 2 should be about materials and methods. I cannot read results without first knowing what has been done. The order of the sections should be: 1. Introduction, 2. Materials and Methods, 3. Results, 4. Discussion, 5. Conclusions.
  • Line 156: Separates ‘of O2-‘, it is written together.
  • Line 157: Separates ‘O2- in’, it is written together.
  • Line 242: Table 1 is not well presented. Everything should be on the same line. The titles of the sections with their corresponding superscripts are well written. It is not clear which are the treatments for each variety of pepper. It is not an appropriate table.

Improve also the supplementary material tables.

  • Line 272: In all tables there is the mg/plant unit format (example). However, in this one you use super index mg plant-1. Please unify the criteria and make all graphs the same.
  • Line 282: The discussion is adequate, but sometimes repeats the same thing many times. It is recommended to revise it and avoid repetitions.
  • Line 411: The ‘experimental dising’ section provides a detailed description of the procedures. However, there could be clearer and more structured wording. It is recommended to organise the steps in paragraphs to improve readability. Also, to clarify the total number of treatments, as this can lead to confusion. It is mentioned that there are 6 combinations, but it says 12 treatments. Mention well the levels and repetitions.
  • Line 443: Describe features of the digital calibrator
  • Line 447: What software was used?
  • Line 447: Why was only the leaf area of a single plant measured? Wouldn't it be more interesting for all of them?
  • Line 461: Under ‘dry matter production’, the paragraph has problems of grammatical structure and clarity, especially in the section on the measurement of root activity. It is recommended to reorganise the information and use more precise wording, following standard scientific language. Also, clarify whether the protocol cited was followed exactly or whether there were modifications.
  • Line 469: What is GR?
  • Line 470: What is GR?
  • Line 471: Rewrite ‘Cool the digestion tnak and rinse the innver cover with a small amount of distilled water, and fill with distelled water to 50 mL’.
  • Line 485: Can you define what a ‘measuremente scheme’ is?
  • Line 488: Can you give more details of the technique used? For example, which column was used.
  • Line 489: Description of the methanol extraction process and subsequent solvent removal is somewhat redundant and unclear. It could be written more concisely.
  • Line 494: What exactly does ‘decolorize’ mean?
  • Line 503: It should add the number of replicas.
  • Line 507: Rewrite ‘The Duncan multiple comparison (LSD test, P<0.05)’ can be confusing.
  • Line 507: Why do you use Duncan's test and not Tukey? Were normality tests performed before ANOVA?
  • Line 510: Conclusions should be better structured. Too much information in a single paragraph. Sentences too long and difficult to understand. No mention of the molecular and physiological mechanisms studied.
  • Line 512: Rewrite sentence ‘effectively solved this problem’ is unobjective and informal.
  • Line 518: Define the term ‘desoprtion’ is the first time it appears in the whole ms. It is in the conclusions but nothing about it is mentioned in the results.

Comments on the Quality of English Language

English needs to be improved, there are grammatical errors.

Author Response

Author's Reply to the Review Report (Reviewer 1)

The main objective of this article is the evaluation of whether exogenous application of selenium (Se) can reduce cadmium (Cd) toxicity in plants of two pepper varieties (one with high and one with low Cd accumulation), grown in a hydroponic system. This is interesting because Cd is a problem in many agricultural lands and the use of Se to mitigate these effects is innovative.

Thus, this ms can be published in this journal after reviewing the next points. In addition to correcting the English, as there are several grammatical errors.

Line 1: Improve the title, the word Cadmium appears too often in the title.

Response: Thank you for your suggestion. We have revised the information of the title in page 1 lines 2-4. Please see the revised version. The statement was list as follow: “Selenium alleviates Cadmium toxicity in pepper (Capsicum annuum L.) by reducing accumulation, enhancing stress resistance, and promoting growth”.

Line 34: Is the most recent regulation from 2014?

Response: Yes, the National Soil Pollution Survey Bulletin was jointly issued by the Ministry of Environmental Protection and the Ministry of Land and Resources of the People's Republic of China in 2014. To date, no updated version of the report has been released."

Line 39: Rewrite ‘Cad is easily transported and accumulates...’.

Response: Thank you for your suggestion. We have rewritten this sentence in page 1-2 lines 44-46. Please see the revised version. The statement was list as follow: “Cd absorbed by crop roots is readily translocated to edible tissues, where it accumulates, posing significant human health risks through dietary exposure”.

Line 64: Unify the naming style of the listed plant species. While most are mentioned by their common name, Coriandrum sativum appears in scientific nomenclature.

Response: Thank you for your suggestion. We have rewritten this sentence in page 2 lines 70-71. Please see the revised version. The statement was list as follow: “Previous reports have shown that the protective effect of Se on crops mainly involves reducing the uptake and accumulation of Cd in some organs of crops or whole plants, such as rice [12], maize (corn) [13], tomato [14], and cilantro (coriander) [15]”.

Line 89: Why do the results appear before the materials and methods? Section 2 should be about materials and methods. I cannot read results without first knowing what has been done. The order of the sections should be: 1. Introduction, 2. Materials and Methods, 3. Results, 4. Discussion, 5. Conclusions.

Response: Thank you for your suggestion. We used the journal template to prepare the manuscript, the order of the sections is: 1. Introduction, 2. Results, 3. Discussion, 4. Materials and Methods, 5. Conclusions.

Line 156: Separates ‘of O2-‘, it is written together.

Response: Thank you for your suggestion. We have revised the information in page 5 lines 163. Please see the revised version.

Line 157: Separates ‘O2- in’, it is written together.

Response: Thank you for your suggestion. We have revised the information in page 5 lines 164. Please see the revised version.

Line 242: Table 1 is not well presented. Everything should be on the same line. The titles of the sections with their corresponding superscripts are well written. It is not clear which are the treatments for each variety of pepper. It is not an appropriate table.

Improve also the supplementary material tables.

Response: Thank you for your suggestion. We have revised Table 1, Table S1 and Table S2 as required. Please see the revised version.

Line 272: In all tables there is the mg/plant unit format (example). However, in this one you use super index mg plant-1. Please unify the criteria and make all graphs the same.

Response: Thank you for your suggestion. We have unified the unit format in Figure 5 and modified it to mg/plant DW. Please see the revised version.

Line 282: The discussion is adequate, but sometimes repeats the same thing many times. It is recommended to revise it and avoid repetitions.

Response: Thank you for your suggestion. We have revised the information in page 9-10 lines 294-298 and page 11-12 lines 391-431. Please see the revised version.

Line 411: The ‘experimental dising’ section provides a detailed description of the procedures. However, there could be clearer and more structured wording. It is recommended to organise the steps in paragraphs to improve readability. Also, to clarify the total number of treatments, as this can lead to confusion. It is mentioned that there are 6 combinations, but it says 12 treatments. Mention well the levels and repetitions.

Response: Thank you for your suggestion. We have described the experimental design in stages, so that we can realize a clearer description. We have revised the information in page 12-43 lines 441, 447 and 462-463. In addition, the experimental design consisted of six treatment combinations, each with three biological replicates, please see the revised version.

Line 443: Describe features of the digital calibrator

Response: Thank you for your suggestion. We have added the description of the digital caliper features. in page 13 lines 477-478. Please see the revised version. The statement was list as follow: “stem base width was measured using a digital vernier caliper (China Ningbo Deli Information Technology Co., Ltd.; accuracy: ±0.2 mm; resolution: 0.1 mm) following standardized protocols”.

Line 447: What software was used?

Response: the root analyzer system was used, We have revised the information in page 13 lines 482. Please see the revised version.

Line 447: Why was only the leaf area of a single plant measured? Wouldn't it be more interesting for all of them?

Response: When using the instrument, only a single plant can be analyzed and determined at a time, and we determined 3 plants per experimental replicate. There may be misunderstanding in the description, so it is revised as follows:“the leaf area measurement system analyzed the total leaf area of three representative seedlings, with three independent biological replicates analyzed per treatment”.It's really interesting to measure all plants, but we still need plants for other indicators. We have revised the information in page 13 lines 483-484. Please see the revised version.

Line 461: Under ‘dry matter production’, the paragraph has problems of grammatical structure and clarity, especially in the section on the measurement of root activity. It is recommended to reorganise the information and use more precise wording, following standard scientific language. Also, clarify whether the protocol cited was followed exactly or whether there were modifications.

Response: Thank you for your suggestion. We have revised the section on the dry matter production and the measurement of root activity in page 13 lines 487-490 and page 14 lines 500-504. Please see the revised version. The statement was list as follow: 

“The root activity of the peppers was determined using the 2,3,5-triphenyltetrazolium chloride (TTC) method as described by Yamauchi et al. [47], with modifications: Root tip samples (0.2-0.3 g) were measured, and the TTC reduction amount was calculated to assess the root dehydrogenase activity”.

“For each treatment, six seedlings were separated into roots, stems, and leaves. The corresponding tissues from all seedlings were pooled to form one composite sample per plant part, with three independent replicate samples per treatment. The samples were first deactivated at 105 °C for 30 min, and then dried to a constant weight at 75 °C before weighing”.

Line 469: What is GR?

Response: Guaranteed Reagent, Belonging to the high-purity chemical reagent grade. We have revised the information in page 14 lines 509. Please see the revised version. The statement was list as follow: “HNO3 (Guaranteed Reagent, GR).

Line 470: What is GR?

Response: GR is abbreviation of Guaranteed Reagent.

Line 471: Rewrite ‘Cool the digestion tnak and rinse the innver cover with a small amount of distilled water, and fill with distelled water to 50 mL’.

Response: Thank you for your suggestion. We have revised the information in page 14 lines 511-513. Please see the revised version. The statement was list as follow: “After cooling the digestion tank and rinsing the inner cover with a minimal volume of distilled water, the digestate was then quantitatively transferred to a 50 mL volumetric flask and brought to volume with distilled water”.

Line 485: Can you define what a ‘measuremente scheme’ is?

Response: Thank you for your suggestion. We have revised the information in page 14 lines 523-527.

Please see the revised version. The statement was list as follow: “The ROS (H2O2, O2-), MDA, Pro, antioxidant enzyme (CAT, POD, SOD), antioxidant substance (ASA/Vitamin C), and IAAO contents were determined with a kit (Suzhou Keming Biotechnology Co., LTD., China) using ultraviolet-visible spectrophotometry according to the manufacturer's standardized protocols. Each treatment was measured three times”.

Line 488: Can you give more details of the technique used? For example, which column was used.

Response: Thank you for your suggestion. We have revised the information in page 14 lines 540-547.

Please see the revised version. The statement was list as follow: “Samples (10 μL) were chromatographed on a Kromasil C18 reversed-phase column (250 × 4.6 mm). The ZR and ABA contents were calculated by measuring the peak area of the standard samples and the samples to be tested after adjusting the mobile phase configuration (1% acetic acid aqueous solution and methanol), flow rate (0.8 mL/min), and column temperature (30 °C). The UV detection wavelength was set at 254 nm, and the sampling duration was 35 minutes. The ABA and ZR standards were purchased from Shanghai Yuanye Biotechnology Co., LTD., China. Three independent biological replicates were analyzed for each sample”.

Line 489: Description of the methanol extraction process and subsequent solvent removal is somewhat redundant and unclear. It could be written more concisely.

Response: Thank you for your suggestion. We have revised the information in page 14 lines 530-533.

Please see the revised version. The statement was list as follow: “Approximately 0.2 g of fresh leaves was weighed and put into 2 mL tubes, and 1 mL of 80% pre-cooled methanol was added to each tube, followed by homogenization at 4 °C for 16 h using an automatic homogenizer and then centrifugation at 8000× g for 10 min to obtain the supernatant”.

Line 494: What exactly does ‘decolorize’ mean?

Response: The decolorization step with petroleum ether serves to remove lipophilic pigments (e.g., chlorophylls, carotenoids) and non-polar interferents that may compromise chromatographic resolution and detection sensitivity.

Line 503: It should add the number of replicas.

Response: Thank you for your suggestion. Three biological replicates were added in  page 15 lines 550.

Line 507: Rewrite ‘The Duncan multiple comparison (LSD test, P<0.05)’ can be confusing.

Response: Thank you for your suggestion. We have revised the information in page 13 lines 553-555.

Please see the revised version. The statement was list as follow: “The results were analyzed by two-way and three-way ANOVA. The mean values of each treatment were subjected to multiple comparisons by the LSD test (P < 0.05)”.

Line 507: Why do you use Duncan's test and not Tukey? Were normality tests performed before ANOVA?

Response: Thank you for raising these important methodological points. After verification, it was confirmed that the LSD test was used in this study, LSD test was restricted to planned comparisons post-ANOVA (P < 0.05) to optimize sensitivity for treatment effects, with assumptions verified. In addition, normality tests were performed before ANOVA.

Line 510: Conclusions should be better structured. Too much information in a single paragraph. Sentences too long and difficult to understand. No mention of the molecular and physiological mechanisms studied.

Response: Thank you for your suggestion. We have rewritten the conclusion in page 15 lines 557-568.

Line 512: Rewrite sentence ‘effectively solved this problem’ is unobjective and informal.

Response: Thank you for your suggestion. We have revised the information in page 15 lines 558-559.

The statement was list as follow: “Se supplementation effectively alleviated Cd toxicity through three mechanisms”.

Line 518: Define the term ‘desoprtion’ is the first time it appears in the whole ms. It is in the conclusions but nothing about it is mentioned in the results.

Response: Thank you for your suggestion. In order to better concise the conclusion, we deleted this sentence.

Reviewer 2 Report

Comments and Suggestions for Authors

This study explores the effect of exogenous selenium (Se) on cadmium (Cd) toxicity in two cultivars of pepper (Yuefeng 750 and Hongtianhu 101) under hydroponic conditions. It evaluates the impact of different Se concentrations on Cd uptake, antioxidant responses, hormone levels, and growth parameters. With major language revision and minor clarifications, this manuscript is suitable for publication in Plants or a related journal.

Major issues:

  • The manuscript suffers from numerous language errors, awkward phrasing, and typos.
  • Some sentences are repetitive and could be more concise.
  • Figures and tables are well-structured, but many references to them in text (e.g., "Fig. S1", "Table S1") are hard to follow without the actual visuals included in the main text.
  • Future directions could include

Abstract

The abbreviations must be be introduced at the first mention of the full name. Abbreviations are typically introduced after the first mention of the word/term in the text.

Line 24: „Meantime, ...“

Introduction :

Line 34-38: The sentence is to long and difficult for understending. Please paraphase.

Line 36: “ ????

Line 36-49: The sentence is to long and difficult for understending. Also it has grammatical errors. Please paraphase.

Line 68: deeply loved???’

Material and methods:

Line 463: New sentence should start with capital letter.

Line 469: What is abbreviation “GR” ?

Line 493: The sentence can not ctart with number.

Line 506: “Origin pro…”

Results:

Line 156: “of O2-

Line 176: What are Pro and ASA content?

Line 186: Abbreviations are typically introduced after the first mention of the word/term in the text.

Line 199: Abbreviations are typically introduced after the first mention of the word/term in the text.

Comments on the Quality of English Language

The manuscript suffers from numerous language errors, awkward phrasing, and typos.

Author Response

Author's Reply to the Review Report (Reviewer 2)

This study explores the effect of exogenous selenium (Se) on cadmium (Cd) toxicity in two cultivars of pepper (Yuefeng 750 and Hongtianhu 101) under hydroponic conditions. It evaluates the impact of different Se concentrations on Cd uptake, antioxidant responses, hormone levels, and growth parameters. With major language revision and minor clarifications, this manuscript is suitable for publication in Plants or a related journal.
Major issues:
The manuscript suffers from numerous language errors, awkward phrasing, and typos.
Response: Thank you for your suggestion. We have polished the English language through MDPI author service (https://www.mdpi.com/authors/english).
Some sentences are repetitive and could be more concise.
Response: Thank you for your suggestion. We have polished the English language through MDPI author service (https://www.mdpi.com/authors/english).
Figures and tables are well-structured, but many references to them in text (e.g., "Fig. S1", "Table S1") are hard to follow without the actual visuals included in the main text.
Response: Thank you for your suggestion. Due to length limitations and the varying significance of the data, some figures and table have been included in the supplementary materials. 
Future directions could include
Abstract
The abbreviations must be be introduced at the first mention of the full name. Abbreviations are typically introduced after the first mention of the word/term in the text.
Response: Thank you for your suggestion. We have revised the information in page 1 lines 22-25. Please see the revised version. 
Line 24: „Meantime, ...“
Response: Thank you for your suggestion. We have changed it to Meanwhile in page 1 lines 27. Please see the revised version. .
Introduction :
 Line 34-38: The sentence is to long and difficult for understending. Please paraphase.
Response: Thank you for your suggestion. First,We have revised the information in page 1-2 lines 37-42. Please see the revised version. The statement was list as follow: “ In 2014, the National Soil Pollution Survey Bulletin issued by the Ministry of Environmental Protection and the Ministry of Land and Resources of the People's Republic of China, showed that the rate of Cd exceedance in cultivated soil (7.0%) ranked first among heavy metals and metalloids, indicating that soil Cd pollution was the most serious in China”.
In short, two government departments in China jointly issued an announcement, which showed the Cd pollution degree of cultivated land soil in China.
Line 36: “ ????
Response: Thank you for your suggestion. We have deleted this redundant punctuation mark in page 1 lines 39.
Line 36-49: The sentence is to long and difficult for understending. Also it has grammatical errors. Please paraphase.
Response: Thank you for your suggestion. We have revised the information in page 2 lines 42-46. Please see the revised version. The statement was list as follow: “ Due to human activities such as mining, fertilization, and pesticide application, both the area affected by heavy metal pollution and its severity in soils continue to increase”.
Line 68: deeply loved???’
Response: Thank you for your suggestion. We have revised the information in page 2 lines 74. Please see the revised version. The statement was list as follow: “ Pepper (Capsicum annuum L.) is widely cultivated and consumed due to its high nutritional and medicinal value”.
Material and methods:
Line 463: New sentence should start with capital letter.
Response: Thank you for your suggestion. We have revised the whole paragraph in page 13 lines 476-480. Please see the revised version. The statement was list as follow: “For each treatment, six seedlings were separated into roots, stems, and leaves. The corresponding tissues from all seedlings were pooled to form one composite sample per plant part, with three independent replicate samples per treatment. The samples were first deactivated at 105 °C for 30 min, and then dried to a constant weight at 75 °C before weighing”.
Line 469: What is abbreviation “GR” ?
Response: Guaranteed Reagent, Belonging to the high-purity chemical reagent grade. We have revised the information in page 14 lines 509. Please see the revised version. The statement was list as follow: “HNO3 (Guaranteed Reagent, GR).
Line 493: The sentence can not ctart with number.
Response: Thank you for your suggestion. We have revised the information in page 14 lines 534-535. Please see the revised version. The statement was list as follow: “ The organic phase in the supernatant was removed by blowing nitrogen at 40 °C”.
Line 506: “Origin pro…”
 Response: Thank you for your suggestion. We have revised the information in page 15 lines 552-553. Please see the revised version. The statement was list as follow: “ and graphed using OriginPro 9.0 (OriginLab Corporation, Northampton, MA, USA).
Results:
Line 156: “of O2-„
Response: Thank you for your suggestion. We have revised the information in page 5 lines 163. Please see the revised version. 
Line 176: What are Pro and ASA content?
Thank you for your suggestion. We have revised the information in page 1 lines 24. Please see the revised version. The statement was list as follow: “and proline (Pro), ascorbic acid (ASA).....”
Line 186: Abbreviations are typically introduced after the first mention of the word/term in the text.
Thank you for your suggestion. We have revised the information in page 6 lines 195-196. Please see the revised version. The statement was list as follow: “Analysis of antioxidative enzyme activities: superoxide dismutase (SOD), catalase (CAT), and peroxidase (POD)”.
Line 199: Abbreviations are typically introduced after the first mention of the word/term in the text.
Thank you for your suggestion. We have revised the information in page 1 lines 22-24. Please see the revised version. The statement was list as follow: “and proline (Pro), ascorbic acid (ASA).....”
